# Structures of Three Actinobacteriophage Capsids: Roles of Symmetry and Accessory Proteins

**DOI:** 10.3390/v12030294

**Published:** 2020-03-08

**Authors:** Jennifer Podgorski, Joshua Calabrese, Lauren Alexandrescu, Deborah Jacobs-Sera, Welkin Pope, Graham Hatfull, Simon White

**Affiliations:** 1Biology/Physics Building, Department of Molecular and Cell Biology, University of Connecticut, 91 North Eagleville Road, Unit-3125. Storrs, CT 06269-3125, USA; jennifer.podgorski@uconn.edu (J.P.); joshua.calabrese@uconn.edu (J.C.); lauren.alexandrescu@uconn.edu (L.A.); 2Clapp Hall, Department of Biological Sciences, University of Pittsburgh, 4249 Fifth Avenue, Pittsburgh, PA 15260, USA; djs@pitt.edu (D.J.-S.); welkin@pitt.edu (W.P.); gfh@pitt.edu (G.H.)

**Keywords:** actinobacteria, actinobacteriophage, bacteriophage, cryoEM, capsid, accessory protein, decoration protein, minor capsid protein

## Abstract

Here, we describe the structure of three actinobacteriophage capsids that infect *Mycobacterium smegmatis*. The capsid structures were resolved to approximately six angstroms, which allowed confirmation that each bacteriophage uses the HK97-fold to form their capsid. One bacteriophage, Rosebush, may have a novel variation of the HK97-fold. Four novel accessory proteins that form the capsid head along with the major capsid protein were identified. Two of the accessory proteins were minor capsid proteins and showed some homology, based on bioinformatic analysis, to the TW1 bacteriophage. The remaining two accessory proteins are decoration proteins that are located on the outside of the capsid and do not resemble any previously described bacteriophage decoration protein. SDS-PAGE and mass spectrometry was used to identify the accessory proteins and bioinformatic analysis of the accessory proteins suggest they are used in many actinobacteriophage capsids.

## 1. Introduction

The *Actinobacteria* are a gram-positive phylum that contains many human pathogens, including *Mycobacterium tuberculosis* and *Mycobacterium abscessus*. A recent renaissance in using bacteriophages to treat multi-drug resistant bacteria has led to a handful of successful cases. The most recent being the treatment of a patient infected with multi-drug resistant *Mycobacterium abscessus* [1]. The viruses (bacteriophages) that infect these bacteria contain a huge reservoir of genes with no known homologues to those found in the domains of life. To characterize these bacteriophages and their genes, the Science Education Alliance Phage Hunters Advancing Genomics and Evolutionary Science (SEA-PHAGES) program advances a course-based research experience for early-career undergraduate students. Currently, there are 145 participating institutions, and the program collectively has isolated over 17,000 bacteriophages of *Actinobacteria* hosts [2,3], of which over 3000 have been sequenced and annotated.

Actinobacteriophages are tailed bacteriophages in the order of *Caudovirales* and are characterized by the virion tail morphology. *Siphoviridae* have a long non-contractile tail, *Podoviridae* have a short non-contractile tail and *Myoviridae* have a contractile tail. Of the 2989 sequenced and annotated actinobacteriophage genomes, 92.9% are *Siphoviridae*, 6.2% are *Myoviridae* and 0.9% are of the *Podoviridae* morphology. This distribution is similar to other structurally characterized bacteriophages that infect a wide range of bacterial hosts [4,5]; whether these ratios reflect the real bacteriophage population or are an artifact from isolation methods is uncertain.

All three morphologies have the capsid head in common, which contains the double stranded DNA genome. All known structurally characterized *Caudovirales* share a conserved fold in their major capsid protein: the HK97-fold, first characterized in the HK97 bacteriophage [6]. The dsDNA in the capsid is tightly packed and creates an internal pressure (10-60 atmospheres) within the capsid [7,8]. Bacteriophages have evolved different mechanisms to strengthen the capsid against this internal pressure. In the HK97 bacteriophage, the major capsid protein forms covalent cross-links with neighboring capsid proteins [6]. Other bacteriophages rely on minor capsid proteins, such as gpD of the *E.coli* lambda bacteriophage [9], which bind to the capsid, stiffen the capsid shell and increase the stability [10]. As well as the major and minor capsid proteins, a number of bacteriophages have been characterized that have decoration proteins bound to their capsid. Their role is unclear; however, the removal of some decoration proteins, for example the Hoc protein of the T4 bacteriophage [11], suggests that they have little to no effect on capsid stability. Structural studies of the Hoc protein revealed an immunoglobulin-like fold, which is typically found in cell attachment proteins, and suggests a possible role in host attachment [12]. Whereas most minor capsid proteins assemble as a trimer, decoration proteins have a wide range of structures, from relatively small dimers [13] to large protrusions [14].

Here, we describe the medium resolution structures of four accessory proteins from three actinobacteriophages identified by students in the SEA-PHAGES program. Using cryo electron microscopy and mass spectrometry, we resolved the capsid heads of one *Myoviridae* and two *Siphoviridae* to approximately six angstroms and identified two minor capsid proteins and two decoration proteins.

## 2. Materials and Methods

### 2.1. Bacteriophages, Hosts, and Sources

The bacteriophages Patience, Rosebush, Myrna and their host, *Mycobacterium smegmatis* mc^2^155 [15], were retrieved from the SEA-PHAGES archive at the University of Pittsburgh. The three bacteriophages were isolated from a number of different countries and environments and have been documented previously (Table 1).

Bacteriophage Patience [16,17] was isolated in Durban, South Africa at the Nelson Mandela School of Medicine (NCBI accession number: JN412589). Relative to the GC content of the host genome (67.4%), Patience has a relatively low genome GC content (50.3%), which has been suggested to mean that it has moved from a different host (with a lower GC content than *M. smegmatis*) relatively recently [18,19]. Patience is one of two members of Cluster U and is lytic. Bacteriophage Rosebush was isolated from the Bronx Zoo, Bronx, NY, USA [19,20,21] (NCBI accession number: AY129334). Rosebush is part of the Cluster B2, which has twenty-seven members, and is lytic. Bacteriophage Myrna [22,23,24,25] was isolated in Upper Saint Clair, PA, USA (NCBI accession number: EU826466). Myrna is one of two members of Subcluster C2 and is lytic.

### 2.2. Production of Bacteriophages for Cryo-Electron Microscopy and Mass Spectrometry

Thirty webbed plates (plates where the plaques are so numerous that they begin to touch) were made for each bacteriophage with 150 mm petri dishes using *M. smegmatis* lawns in top agar (per liter: 4.7 g 7H9, 3.5 g BactoAgar, 0.1 *v/v* glycerol and 1 mM CaCl_2_ final) on Luria agar plates (per liter: 15.5 g Luria broth base, 15 g Agar with the final concentrations of 50 ug/mL carbenicillin and 10 ug/mL cycloheximide). To 600 µL of stationary phase *M. smegmatis* (grown for 4 days at 37 °C at 250 rpm in Middlebrook 7H9 liquid medium) enough bacteriophages were added to ensure a webbed plate (note: this was empirically determined with each batch of bacteriophage). Bacteriophages and bacteria were incubated at room temperature for 20 min to allow infection. Ten milliliters of top agar was then added and quickly poured onto agar plates. Patience and Rosebush were incubated at 37 °C while Myrna was incubated at 30 °C. Plates were incubated for 36–48 h to allow optimal plaque formation. Then, 10 mL of Phage Buffer (10 mM Tris-HCl pH 7.5, 10 mM MgSO_4_, 68 mM NaCl, 1 mM CaCl_2_) was added to each plate and incubated for 4 h at room temperature to create a bacteriophage lysate. The lysate was then aspirated from the plates and pooled together before being centrifuged at 5500× *g* for 10 min at 4 °C. Bacteriophage particles were precipitated with the addition of NaCl (1 M final concentration) and PEG-8000 (10% *w/v* final concentration) and mixed overnight at 4 °C. The bacteriophage particles were pelleted by centrifugation at 5500× *g* at 4 °C for 10 min and the supernatant discarded. The bacteriophage particles in the pellet were then resuspended in 10 mL of Phage Buffer by gentle rocking overnight at 4 °C. The new bacteriophage lysate was then centrifuged at 5500× *g* for 10 min to remove any debris, and 8.5 g of CsCl was added to the 10 mL of bacteriophage lysate for a final density of 1.5 g CsCl/mL of bacteriophage lystate. The CsCl/bacteriophage solutions were then centrifuged at 192,068× *g* for 16 h and the bacteriophage particle band removed with a syringe and needle. Bacteriophage particles were then dialyzed against Phage Buffer and pelleted again to concentrate the bacteriophage particles before being resuspended in 50 µL of Phage Buffer. Purified bacteriophage particles were stored at −80 °C with dimethyl sulfoxide (6% final concentration).

### 2.3. Preparation of Bacteriophages for Transmission Electron Microscopy

Twenty microliters of each bacteriophage from −80 °C storage (concentration unknown) was dialyzed into Phage Buffer using a Tube-O-Dialyzer Micro (G-Biosciences, St Louis, MO, USA) with a 50 kDa molecular weight cut-off. After this, 3 µL of dialyzed bacteriophage particles was used for uranyl acetate negative stain electron microscopy using a Tecnai G2 200 keV transmission electron microscope (FEI, Hilsboro, OR, USA).

### 2.4. Preparation of Cryo-Electron Microscopy Grids

Five microliters of Patience and Rosebush were mixed together in a 1.5 mL tube, as were 5 µL of Myrna and Rosebush in a different 1.5 mL tube. These mixtures are referred to as the multiplexed bacteriophage. Five microliters of multiplexed bacteriophages were pipetted onto a C-flat 2/1-2C (2 µm hole, 1 µm space) cryo-electron microscopy grid (Protochips, Morrisville, NC, USA) using a Vitrobot mk IV (FEI, Hilsboro, OR, USA). Grids were blotted for 5 s with a force of 5 (a setting on the Vitrobot) before being plunged into liquid ethane.

### 2.5. Cryo-Electron Microscopy

Data were collected on a 200 keV Talos Artica (FEI, Hilsboro, OR, USA) at the University of Massachusetts medical school equipped with a K3 direct electron detector (Gatan, Pleasanton, CA, USA). In total, 996 movies were taken for the Patience and Rosebush grid while 1200 images were taken for the Rosebush and Myrna grid. Movies were taken at 36000x magnification (1.1 Å pixel size) and consisted of 32 frames, each with a dose of 0.550074 e^−^/Å^2^ (total exposure time per movie was 1.59787). A defocus range of −0.5 to −2.5 µm was used with 0.5 µm steps.

### 2.6. Cryo-Electron Microscopy Data Analysis

Relion 3.0.4 [26] was used for bacteriophage capsid reconstructions. Data were 2x binned so that the pixel size was 2.2 Å to speed up computation. The box size was 400 pixels (880 Å) with a mask diameter of 800 Å. A total of 1000 bacteriophage particles were manually picked for 2D classification to create references for the subsequent auto-picking of bacteriophage particles using Relion. After extraction, the particles were subjected to 2D and 3D classification with the 2D classes used to create the 3D initial model using the Relion software. The different capsids were clearly separated into different 3D classes. Each class was used separately for refinement using I1 icosahedral symmetry. Masks were created using the Relion software and used in the final post-processing. Contrast Transfer Function refinement and Bayesian polishing were not performed. Figures for this paper were prepared using Chimera [27] and ChimeraX [28]. The Segger [29] function in Chimera was used to segment out the different proteins.

#### 2.6.1. Data Deposition

Electron Microscopy Data Bank accession numbers are as follows: Rosebush: EMD-21122, Patience: EMD-21123, Myrna: EMD-21124.

### 2.7. Bioinformatics

The ITASSER [30,31], HHPRED [32], BLAST-P and PSI-BLAST [33] online servers were used with the default settings. All four online servers were used to investigate the four accessory proteins and the Rosebush capsid protein (gp15). ITASSER was used for protein structure prediction, HHPRED and PSI-BLAST for structural homology matching and BLAST-P for amino acid sequence identity comparisons. The following databases were used in HHPRED: PDB_mmCIF70_28_Nov, PDB_mmCIF30_28_Nov, SCOPe70_2.07, ECOD_ECOD_F70_20190225, COF_KOG_V1.0, Pfam-A_v32.0, NCBI_Conserved_Domains(CD)_v3.16, SMART_v6.0, TIGRFAMS_v15.0 and PRK_v6.9. For BLAST-P and PSI-BLAST, the Non-redundant protein sequences (nr) database was used. Alignments of bacteriophage proteins were done using Clustal Omega [34].

### 2.8. Mass Spectrometry

#### 2.8.1. In-Gel Digestions, Tryptic Peptide Extraction and Desalting

Concentrated bacteriophage particles were boiled in 2x SDS-PAGE loading buffer and run on a 10% denaturing SDS-PAGE alongside a Precision Plus dual color protein ladder (Bio-Rad, Hercules, CA, USA). After Coomassie staining, bands were excised and destained using 40% ethanol, 10% acetic acid in water. The gel bands were diced, equilibrated to pH 8 in 0.1 M ammonium bicarbonate, and subjected to Cys reduction and alkylation using 10 mM dithiothretol in 0.1 M ammonium bicarbonate (1 hr at 37 °C) and 55 mM iodoacetamide in 0.1 M ammonium bicarbonate (45 min at 37 °C in the dark), respectively. The gel bands were dehydrated using acetonitrile, dried to completion in a Speedvac concentrator (Labconco, Kansas City, MO, USA) and resuspended in a 12.5 ng/µL trypsin solution in 0.1 M ammonium bicarbonate for 45 min on ice. The supernatant was removed and replaced with 0.1 M ammonium bicarbonate. Proteolysis continued for 16 hr at 37 °C on a thermal mixer (Eppendorf, Hamburg, Germany). Tryptic peptides were extracted using consecutive hydration and dehydrated using 0.1 M ammonium bicarbonate and 50% acetonitrile in 5% formic acid. Two final hydration and dehydration cycles were conducted using 0.1 M ammonium bicarbonate and 100% acetonitrile. Pooled peptide solutions were dried in a Speedvac concentrator and resuspended in 0.1% formic acid in water. Peptides were desalted using Pierce C18 desalting spin columns (Thermofisher Scientific, Waltham, MA, USA) according to the manufacturer’s instructions.

#### 2.8.2. Peptide and Protein Identification Using Mass Spectrometry

Dried and desalted peptides were resuspended in 0.1% formic acid in water and analyzed using nanoflow ultra-high-performance liquid chromatography (UPLC) coupled to tandem mass spectrometry (MS/MS) using a Dionex Ultimate 3000 RSLCnano UPLC system and Q Exactive HF mass spectrometer (Thermofisher Scientific, Waltham, MA, USA). Peptides were separated using a 1 hr reversed-phase UPLC gradient over a 75 µm × 25 cm Easy Spray PepMap C18 analytical column (Thermofisher Scientific, Waltham, MA, USA) and directly ionized into the Q Exactive HF using positive mode electrospray ionization. The UPLC gradient implemented a 300 nL/min flow rate and the following Solvent A (0.1% formic acid in water) and Solvent B (0.1% formic acid in acetonitrile) conditions: initial 4% Solvent B for 10 min, followed by a linear ramp to 30% Solvent B over 40 min, then a linear ramp to 90% Solvent B over 10 min. Finally, a 10 min 90% Solvent B wash was carried out, followed by a 2 min linear ramp to initial conditions and an 18 min re-equilibration. A high resolution MS1 and data-dependent Top15 MS/MS acquisition method was used. All raw data were searched against the Uniprot *Mycobacterium* baceriophage Rosebush reference proteome (UP000006817) using MaxQuant v1.6.0.1 ([35], Planegg, Germany) and default parameters plus the following additional settings: N-terminal Gln to pyro-Glu variable modification and minimum 5 amino acids/peptide. Identifications were filtered to a peptide- and protein-level false discovery rate of 1% using a decoy search and uploaded into Scaffold 4.10.0 (Proteome Software, Portland, OR, USA) for visualization and further analysis.

## 3. Results

### 3.1. Virion Morphologies

All the viruses described here are double stranded DNA tailed bacteriophages of the viral order *Caudovirales* (Figure 1). Patience and Rosebush show the *Siphoviridae* morphology (long non-contractile tails), while Myrna has *Myoviridae* morphology (contractile tail).

### 3.2. Capsid Morphologies and Accessory Proteins

The three bacteriophages were multiplexed onto two grids with Patience and Rosebush on one grid and Myrna and Rosebush on another (Figure 2). The rationale was that this method would allow for more structures from less microscope time. We hypothesized that the Relion software would separate the particles at either the 2D or 3D classification stage. All the particles were picked with no attempt to choose specific bacteriophage particles, for example either just Patience or just Rosebush. Two-dimensional classification had some success in separating out the different bacteriophage particles, but it was not clear cut, with some ambiguous orientations of phage particles. Therefore, all “good” 2D classes (Appendix A)—those with very clear features in the 2D class images—were selected and used in 3D initial model generation and 3D classification. It was clear in the 3D classes that the bacteriophage particles had been separated successfully, with the decoration proteins (described later) of Patience and Rosebush clearly recognizable. To rule out chimeric reconstructions, for example where Rosebush and Patience particles are used to make the cryo-EM map, Rosebush was added to both multiplexed grids. In both cases, Relion classified the Rosebush particles separately and the final map was identical. The three bacteriophage structures were successfully resolved from the datasets (Figure 3) at the medium resolutions of 7.7 Å (Myrna), 5.9 Å (Patience), and 6.7Å (Rosebush), based on the reference-based Fourier shell correlation criterion (FSC = 0.143 FSC). We did observe that a minority of bacteriophage particles had released their DNA (Appendix A, third row, second column, shows the empty particles). This may be a result of thawing the bacteriophages from −80 °C.

Patience shows pseudo T = 7 laevo capsid organization, as is common among many dsDNA bacteriophages. Regarding the newly proposed framework to describe capsid organization, it appears to have T_t_(2,1) = 28/3 (~9.33, which means it has a surface area close to the classic Caspar and Klug T = 9) organization [36] and has a minor coat protein that surrounds the major capsid protein hexamer (Figure 4). Patience also has a decoration protein that links the minor coat proteins together and makes no contact with the major capsid protein (Figure 5). The major capsid proteins that make up the hexamers (Figure 4) have an arrangement that appears similar to the 80alpha bacteriophage procapsid (EMD-7030) [37]. The internal diameter of the capsid is 63.5 nm.

Previous work characterized the proteins in the mature capsid of Patience using mass spectrometry [18] and SDS-PAGE. Their SDS-PAGE analysis showed three prominent bands that matched the estimated (Protparam software [38]) molecular weights of the capsid subunit (gp23), major tail subunit (gp31) and a protein with unknown function (gp15). The mass spectrometry analysis of the whole capsid identified the capsid subunit (gp23), gp15, gp4, gp29 and gp31.

Based on these previous results, we considered gp4, gp15 and gp29 (Figure 6) as candidates for the minor capsid and decoration protein based on the number of expected copies of each protein in the mature capsid. All three proteins had their structure predicted using the I-TASSER server [30,31] to enable fitting into the cryoEM density. The position of the gp4 gene, outside the expected syntenic gene order of structural proteins, and small size (102 amino acids, 10.5 kDa) in the genome makes it unlikely to be part of the structural proteins. The ITASSER predicted structure of gp29 (139 amino acids, 15 kDa) fits the density of the decoration protein well but the C-score (−3.78) and TM-score (0.3 ± 0.1) suggest that the prediction is not that accurate. However, its position in the genome, and proximity to the major capsid protein (gp23) makes this protein most likely to be the decoration protein. The cryo-EM map, when fitted with the ITASSER model (Appendix A), suggests that gp29 exists as a dimer in the capsid. With respect to the minor capsid protein, gp15 is the only remaining candidate based on the previous SDS-PAGE and mass spectrometry analysis. However, the I-TASSER model does not resemble the density, nor do HHPRED [32] and PSI-BLAST [33] predict any structural homology for any other known proteins.

Rosebush shows T = 9 capsid organization. In the newly proposed framework, this equates to T_h_(3,0) = 9. It has a decoration protein that forms raised hexamers that sit directly above the capsid hexamers and makes contact with the major capsid proteins beneath. It has a similar internal capsid diameter to Patience (61.4 nm). The major capsid protein shows HK97-fold like properties, with the A domain and E-loop visible. However, HHPRED and PSI-BLAST fail to predict the HK97-fold (Appendix A), as does the I-TASSER software (C-score range of −5 to −3.36 and TM-score of 0.34), suggesting that this may be a novel variant of the HK97-fold.

HHPRED and PSI-BLAST analysis of gp17 (which we propose to be the decoration protein) shows no significant matches to any known protein structure. The amino acid sequence was submitted to the I-TASSER protein structure prediction software, but the resulting models had C-scores (−5 to −2.58) and TM-scores (0.29) which suggested a poor predicted model (the best model had a C-score of −3.98 and a TM-score of 0.29 ± 0.09). Comparison of the models to the cryo-EM map showed a poor fit. We also submitted gp17 to the online Robetta server, which also failed to predict an HK97 fold.

Myrna shows T = 16 capsid organization, with a minor capsid protein positioned at each of the six sides of the major capsid hexamer. Following the new framework, it is a T_t_ (4,0) = 64/3 (~21.33, which means it has a surface area close to the classic Caspar and Klug T = 21), similar to HSV-1 [39]. The interior capsid diameter is 81.1 nm. The minor capsid protein gene (*98*) could be identified easily using bioinformatic analysis and sits directly upstream (Figure 6) of the major capsid protein gene (*99*), following the typical gene synteny found in *Caudovirales*. HHPRED suggests that gp98 has structural homology (85.21% probability) with the TW1 [40] bacteriophage (which infects *Pseudoalteromonas phenolica*, a gram-negative marine bacterium [41]), despite only having 20% amino acid sequence identity.

Based on amino acid sequence identity (Appendix A), a related gene can be found in Phrappuccino (gp 54, AA cluster which has two members, host: *M. smegmatis*), 130 bacteriophages (91.5% of the total number of bacteriophages in this cluster) in the C1 subcluster (*M. smegmatis*), Pupper (gp 95), Skog (gp 158) and SCentae (gp 94) of the DO subcluster, which has three members (*G. terrae*), RedWattleHog (gp 15) and Stormageddon (gp 14) of the DX subcluster, which has two members (*G. terrae*), as well as the singletons Finch (gp 18, *R. erthropolis*) and E3 (gp 81, *Rhodococcus equi* NCIMB 10027), all of which have *Myoviridae* morphology. The other *Myoviridae* clusters, DQ (*G. terrae* 3612), EA5 (*M. foliorum*), AR (*Arthrobacter*), AO1 (*Arthrobacter*), and AO2 (*Arthrobacter*), do not have any protein with significant amino acid sequence identity to gp98. However, accessory proteins have been annotated in these bacteriophage; for example, protein 11 in Chipper1996 is predicted by HHPRED to also have structural similarity to the TW1 minor coat protein, suggesting a conserved fold in these minor coat proteins.

### 3.3. SDS-PAGE Gel and Mass Spectrometry Analysis of Rosebush

Purified Rosebush was run on an SDS-PAGE and the four darkest bands were excised for mass spectrometry analysis (Figure 7 and Appendix A). The rationale for this was that the accessory protein should be present in relatively high amounts and produce a dark band on the gel. One of the bands from the Rosebush gel (band 4 in Figure 7) was identified by mass spectrometry analysis to be gp17 (Figure 6). We propose that gp17 is the decoration protein.

### 3.4. Bioinformatic Analysis of Accessory Protein gp17

BLAST-P [33] of the amino acid sequence of gp17 against the non-redundant protein sequences database shows that gp17 is found in a number of other actinobacteriophages in Cluster B. Those phages in the B2 sub-cluster have almost 100% amino acid sequence identity to gp17, while those phages in other B sub-clusters (B1–B13) have far lower sequence identity; for example, phage Phelemich [16] in the B5 sub-cluster has a 31.58% amino acid sequence identity. Partial matches can also be found in the DR cluster (6 members); for example, gp15 of CloverMinnie, which infects the *G. terrae* host. TPA2 (a singleton that infects *Tsukamurella paurometabola* Tpau37) and Anderson (from the UNK subcluster, *M. smegmatis*) also show a partial match (Appendix A).

## 4. Discussion

The bacteriophage structures described here expand our structural knowledge of bacteriophages and their diversity. The decoration proteins of Patience (gp29) and Rosebush (gp17) are novel and have no known structural homologues. Their purpose is unknown and further experiments are needed, for example deletion of the decoration proteins from the bacteriophages, to measure their effect on capsid stability and infectivity. We hypothesize that both accessory proteins of Patience (which we annotate as gp15 and gp29) are likely involved in capsid stability with gp29 acting like other minor coat proteins and gp15 linking them together. The decoration protein gp15 appears to exist as a dimer and the linking of two minor coat proteins with a decoration protein has not been observed in other bacteriophage capsids. The decoration proteins of *Bordetella* infecting bacteriophage, BPP-1 [43], epsilon15 [44] and phage P4 [13], also form dimers on the capsid. However, in those cases, the dimers are located in similar positions to minor capsid proteins and interact with the major capsid protein directly. The role of gp17 in Rosebush is likely to have a minimal effect on structural stability based on the observation that it extends far from the bacteriophage capsid and makes few contacts with the coat protein hexamer underneath. It may play a role in host recognition, making weak contacts with the host before the tail contacts its receptor. Enhancement of capsid attachment to the host has been observed with the ϕ29 bacteriophage head fibers [45], although the fibers are not essential for phage infection [46]. However, it must be pointed out that decoration proteins in other bacteriophages, such as Hoc and Soc in the T4 bacteriophage, have been shown to aid in capsid stability [11]. Therefore, it may be that gp17 of Rosebush does have a role in capsid stability or some other unknown function.

Patience and Rosebush are excellent examples of two of the main mechanisms by which bacteriophages can increase the size of their capsid while using the same HK97-fold building block. In this context, we are describing how a virus with T = 7 capsid architecture might increase the capsid size. Both Patience and Rosebush have similar capsid sizes (63.5 and 61.4 nm internal diameter, respectively) but different T numbers. Rosebush has altered its major capsid protein, and presumably the scaffolding protein that these bacteriophages use in the capsid assembly process. The altered major capsid protein is used to increase the T number from 7 to 9. This change in T number causes the number of major capsid proteins to increase from 420 copies to 540, resulting in a larger capsid. Patience has used a completely different method to increase capsid size by using the same number of major capsid proteins (420 copies in a T = 7 capsid), but adding minor capsid proteins to increase the capsid size. The minor capsid protein of Patience may also play a role in stability in a similar way to the minor capsid protein gpD in bacteriophage lambda [47]. In bacteriophage lambda, gpD exists as a trimer in the capsid and is found at the three fold axes of the mature capsid (gpD is not present in the immature capsid) [9]. Atomic force microscopy experiments have shown that binding of gpD to the immature capsid increases the stiffness of the entire capsid [10] and structural studies suggest gpD replaces the need for the major capsid protein cross-link found in HK97 [6,9].

Structural characterization of T = 9 viral capsids is relatively rare with only Basilisk (*Bacillus cereus*, 18 Å resolution) [47], Ty3/Gypsys retrotransposon capsid (7.5 Å resolution, 6R24) [48], *H. ochraceum* microcompartment shell (3 Å resolution, 6MZX) [49,50], and N4 (14 Å resolution) [51] having published structures, all at a lower resolution than is described here for Rosebush. Rosebush’s coat protein configuration appears to be completely different to other bacteriophages. Further structural prediction and comparison to known databases reveals that the Rosebush major capsid protein may be a novel variant of the HK97-fold. Further work is needed to obtain a higher-resolution map to allow for amino acid model building. NMR [52], X-ray crystallography [6,53] and cryo-electron microscopy [40] have been used previously to determine high resolution protein structures of capsids and accessory proteins and are potential routes for future structural studies of the capsid proteins.

The structure of Myrna shows how common the T = 16 capsid architecture is, with examples of viruses found in all domains of life that have a similar structure, for example the human virus HSV-1 [39]. This is the first time it has been described in the *Actinobacteria*, and suggests that this capsid architecture is adaptable to a number of different hosts and environments to encapsulate large genomes.

Finally, we have shown that the use of multiplexing bacteriophage virions is a successful strategy for lowering the cost per structure. Mixing different bacteriophage virions together allows for a single data collection session on expensive cryo-electron microscopes. We predict that up to five different bacteriophage virions could be easily multiplexed on a single grid, further decreasing the cost. This is an important tool for viral structural studies. There are over 100 sub-clusters of actinobacteriophages alone. In order to quickly, and relatively cheaply, screen a bacteriophage from each sub-cluster, we believe that multiplexing is the best method. Once all the bacteriophages have been screened at 5–10 angstrom resolution, which allows for the HK97-fold to be identified, they can be classified into structural groups and exemplar bacteriophages identified for high resolution structural studies. At the 5–10 angstrom resolution range (so called medium resolution), secondary structures, for example alpha helices, are visible, but tracing the backbone of the protein chain is very challenging [54]. For most protein folds, for example the common double jelly roll fold found in PRD1 [55], medium resolution cryo-EM maps should have visible secondary structures within the folds and so allow for identification and classification of the viral capsids. This method could be widely used for bacteriophages that infect various phyla to rapidly expand the structural capsid database to allow comparison of different bacteriophages. This will have the greatest impact on viral evolution studies. It has long been discussed in the literature that the major capsid protein may be the best way to explore the evolution of viruses [56,57,58]. However, the conservation of the protein may only be found in the protein fold and not in the amino acid sequence. Therefore, collecting many bacteriophage structures is the only way to carry out this evolutionary work.

## 5. Conclusions

We have identified four head accessory proteins in the actinobacteriophages—two are novel decoration proteins and two are minor capsid proteins that show structural homology to known minor capsid proteins in other viruses.

## Figures and Tables

**Figure 1 viruses-12-00294-f001:**
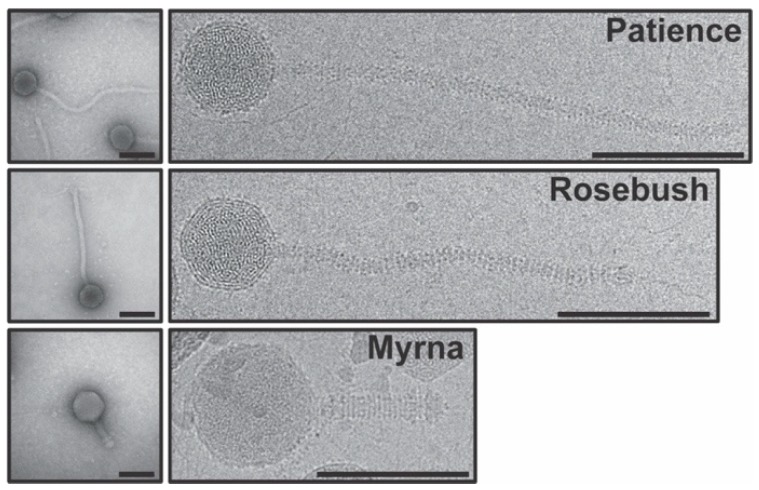
Electron microscopy (EM) of the actinobacteriophages. On the left are representative negative stain images of the three actinobacteriophages, Patience, Rosebush and Myrna, while the right shows representative cryo-EM images. Scale bar in the micrographs is 100 nm. The cryo-EM images are all at the same magnification (36,000×) and their sizes are directly comparable to one another. Bacteriophage dimensions are as follows. Patience head diameter: 66.7 nm, Patience tail length: 330.1 nm. Rosebush head diameter: 66.1 nm, Rosebush tail length: 252.5 nm, Myrna head diameter: 84.8 nm, Myrna tail length: 90.9 nm.

**Figure 2 viruses-12-00294-f002:**
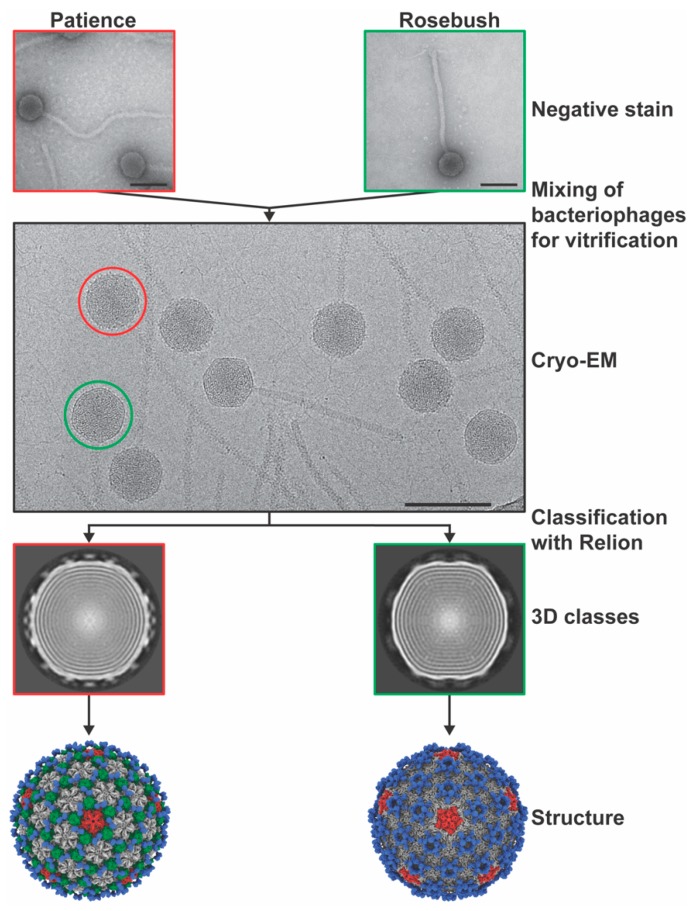
Multiplexing bacteriophage samples. Diagram showing the basic method by which the bacteriophages were multiplexed on a single cryo-EM grid and used for data analysis. Scale bar is 100 nm.

**Figure 3 viruses-12-00294-f003:**
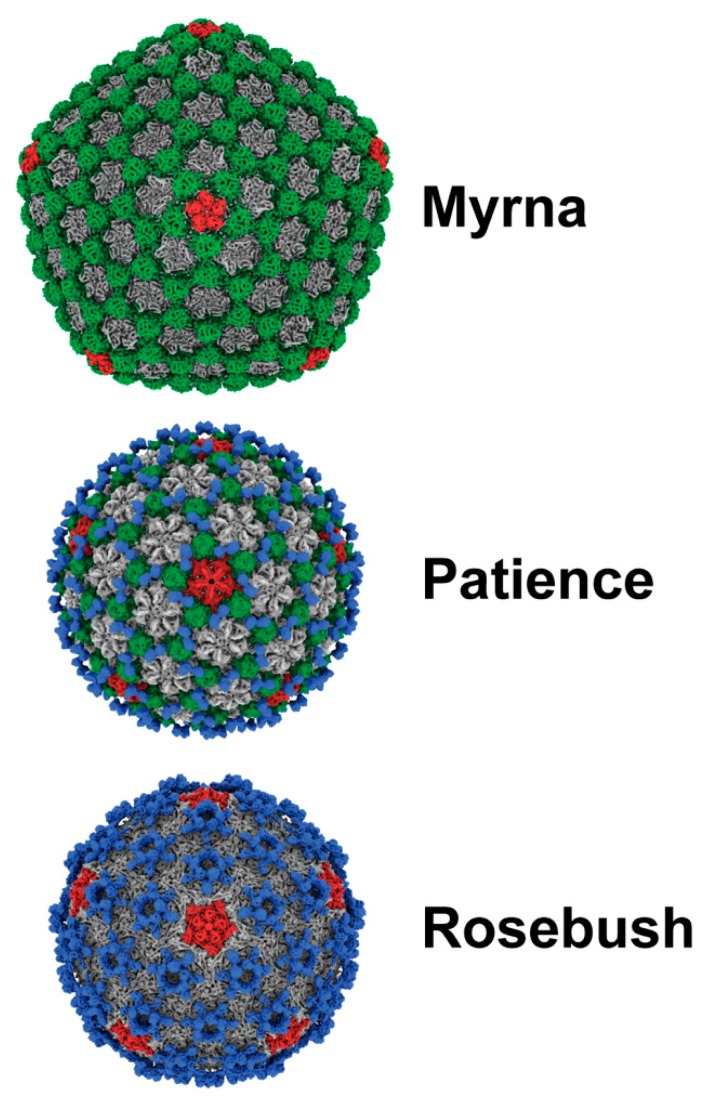
Capsid structure of the actinobacteriophages. Myrna (7.7 Å resolution from 2268 particles), Patience (5.9 Å resolution from 3461 particles) and Rosebush (6.7 Å from 3489 particles) cryo-EM maps. Pentamers are colored red, minor capsid proteins green and decoration proteins blue. Major capsid protein hexamers are grey in color. Cryo-EM maps in this figure and following figures were displayed using the ChimeraX software [28]. EMDB accession numbers are as follows: Rosebush: EMD-21122, Patience: EMD-21123, Myrna: EMD-21124.

**Figure 4 viruses-12-00294-f004:**
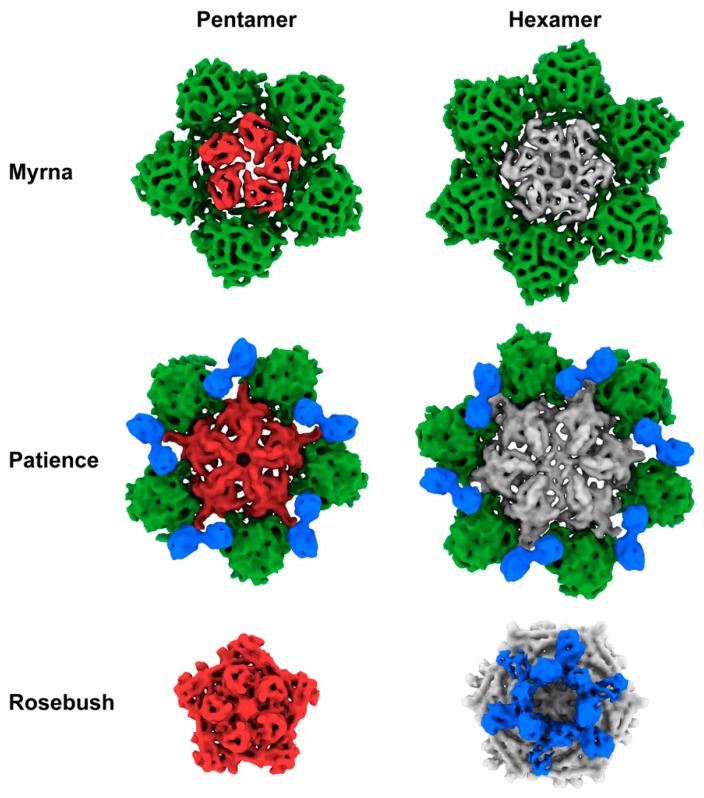
Hexamers and pentamers of the actinobacteriophages. The pentamers and hexamers (including minor/decoration proteins) have been segmented out from the capsid maps using the Segger [29] setting in Chimera [27]. Major capsid proteins are colored grey (hexamer) or red (pentamer). Minor capsid proteins green and decoration proteins blue. Images have not been re-scaled and are comparable.

**Figure 5 viruses-12-00294-f005:**
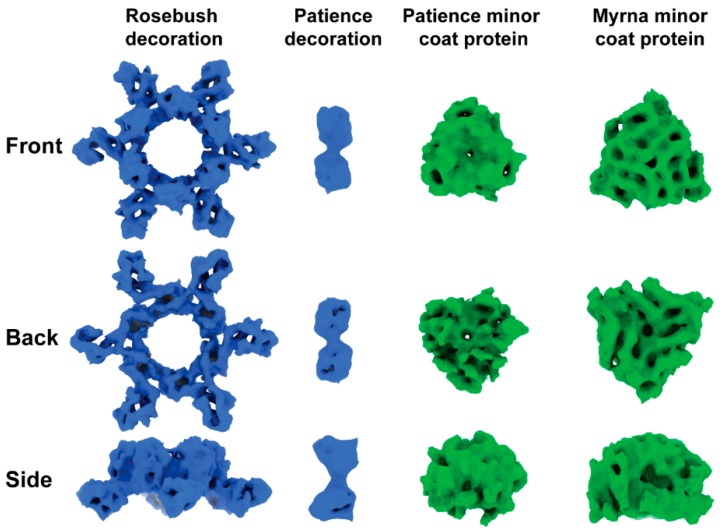
Density of segmented auxiliary proteins. Decoration proteins and minor protein density segmented from capsid maps using the Segger [29] setting in Chimera [27]. Front view is from the outside of the capsid looking in. Back view is from the inside of the capsid looking out. Densities have been scaled relative to one another and their sizes are comparable.

**Figure 6 viruses-12-00294-f006:**
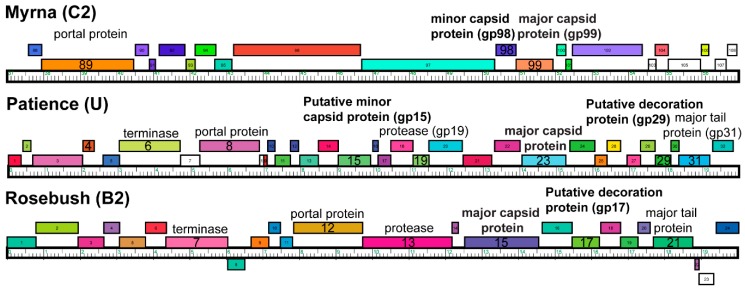
Gene organization of structure proteins. A portion of each bacteriophage genome displayed with the phamerator software [42] showing the positions of the annotated structural proteins.

**Figure 7 viruses-12-00294-f007:**
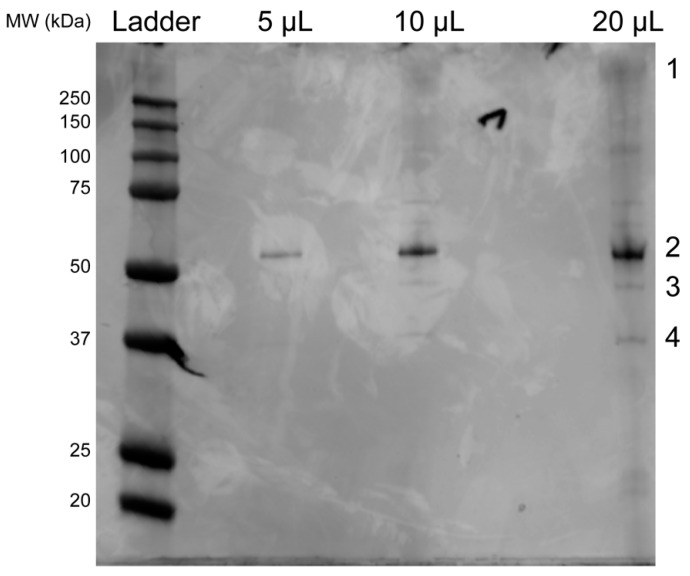
Mass spectrometry identification of accessory proteins in Rosebush. SDS-PAGE gel of Rosebush (top) with the bands excised for mass spectrometry highlighted by 1–4. Bottom: the results of the mass spectrometry shown for each band highlighting the Rosebush proteins with at least 15% sequence coverage. Full mass spectrometry results can be found in Appendix A.

**Table 1 viruses-12-00294-t001:** Basic information about the three actinobacteriophages Myrna, Patience and Rosebush.

Name	Number of Genes	Cluster	Number of Phages in Cluster	Genome Length (bp)	Genome GC Content (%)	Life Cycle	Host	Morphology
Myrna	229	C2	2	164602	65.4	Lytic	*M. smegmatis* mc^2^155	*Myoviridae*
Patience	109	U	2	70506	50.3	Lytic	*M. smegmatis* mc^2^155	*Siphoviridae*
Rosebush	90	B2	27	67480	68.9	Lytic	*M. smegmatis* mc^2^155	*Siphoviridae*

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
