# Peer review of "Structures of Three Actinobacteriophage Capsids: Roles of Symmetry and Accessory Proteins"

_viruses, 2020, doi:10.3390/v12030294_

Round 1
Reviewer 1 Report
This paper reveals 3 novel structures of capsids of 3 actinophages. They identify structural proteins from them identify homologues in related phages. By solving the structures, they are able to thread the sturctural rpteins to give them functional annotataions, thus increasing our understanding of virus structure. It is of high interest to those studying viral diversity and structural biology.
Some minor points:
Minor issues dealing with repeatability of data:
Line 222: How were "good" 2D classes defined? This should be explained.
Minor issues dealing with clarity:
Line 259: What is the Capsid organization number using the new framework (Tt(3,0)?)
Line 230: What plural form of "bacteriophage" are the authors subscribing to? I don't care which one they use, but they refer to multiple bacteriophages in many of the figure legends and other parts of the paper as "bacteriophages". Does the plural form designate a specific type of multiple (such as phages describe multiple speceies whereas phage describes multiple particles of the same species)? This should be consistent.
Lines 277 - 282: The fact that related genes to gp98 can be foudn in AA, C1 and DO is very intriguing, and it would be very informative to have a figure that displays a multiple sequence alignment of these genes. This would allow readers of the paper to understand how these genes are conserved.
Lines 296-300: Once again, this is a very interesting finding, but it is hard to really interpret without a multiple sequence alignment or phylogenetic tree to show the relationshiips. For example, the authors describe that the DR cluster has a partial match, but that isn't an informative statement, because it is unclear what about the match is partial. This ambiguity could be cleared up by showing a multiple sequence alignment.
Lines 370-376: I do not understand this point. The authors describe that Patience and Rosebush have increased their capsid size, but compared to what? They have about the same capsid sizes when compared to each other based on the internal diameter, so they haven't increased their capsid size relative to eachother, so to which phage are the authors comparing these phages to?
Minor issues about the interpretability of the data by the broader community:
Results Section 3.1: This is a very minor point, but this section reiterates known data (and cites it appropriately); to some, it might appear a bit disingenuous to place this section in the Results rather than the Background.
Lines 368-369: The speculation that the decoration proteins are important for host cell interaction is intriguing, and this statement should cite other instances where such capsid associated proteins were useful in binding the host. However, I think it is also important to list other possible functions for these proteins that is also consistent with what is known about decoration proteins in phages, as not to create the idea that binding to the host is their definitive function. While the authors do say that this function is speculative, people unfamiliar with the field and less scrupulous, might interpret that statement as fact and thus propagate the idea that structural evidence shows that decoration proteins are important for host binding. Having alternative hypotheses on what these proteins might bind would help prevent such confusion.
Lines 403-405: The authors use this paragraph how the multiplexing of phage caspids is useful for rapidly multiplexing and deciphering viral structures. They describe that at 6 angstrom resolution, the HK97 fold is decipherable and therefore useful in identifying structural groups. This information is highly useful for the field as a technical benchmark to achieve, but the information is only useful to those who study the structure of Caudovirales. However, other phages do not use this structure, such as the Cortico viridae and Tectiviridae that have a PRD1 like capsid, the Cystoviridae that are BTV-like and folds in things like Leviviridae and Inoviruses have completely novel capsid structures as well. If the authors wish to claim that multiplexing phage capsids is a successful strategy for lowering the cost per structure, a discussion of how this would be useful for other families of phages should be included, as well as a speculation about what level of resolution would be necessary to resolve them. Otherwise, this method is solely useful for Caudovirales phages, which is useful, but should be specified as such.
Author Response
Reviewer 1:
This paper reveals 3 novel structures of capsids of 3 actinophages. They identify structural proteins from them identify homologues in related phages. By solving the structures, they are able to thread the sturctural rpteins to give them functional annotataions, thus increasing our understanding of virus structure. It is of high interest to those studying viral diversity and structural biology.
Some minor points:
Minor issues dealing with repeatability of data:
Line 222: How were "good" 2D classes defined? This should be explained.
Picking of 2D classes is subjective. As stated in the text, where very clear features in the 2D class images could be seen, they were selected. We have added a supporting information figure (S2) for Rosebush and Patience to show the difference between “good” and “bad” images.
Minor issues dealing with clarity:
Line 259: What is the Capsid organization number using the new framework (Tt(3,0)?)
Added “Regarding the newly proposed framework this equates to Th(3,0)=9.”
Line 230: What plural form of "bacteriophage" are the authors subscribing to? I don't care which one they use, but they refer to multiple bacteriophages in many of the figure legends and other parts of the paper as "bacteriophages". Does the plural form designate a specific type of multiple (such as phages describe multiple speceies whereas phage describes multiple particles of the same species)? This should be consistent.
Have attempted to be more consistent. Have used “bacteriophage particle” where possible.
Lines 277 - 282: The fact that related genes to gp98 can be foudn in AA, C1 and DO is very intriguing, and it would be very informative to have a figure that displays a multiple sequence alignment of these genes. This would allow readers of the paper to understand how these genes are conserved.
Have added the multiple sequence alignment for a bacteriophage from each of the clusters. Two singletons and bacteriophage from DX have also been identified as being related and so added to the text and Figure S3.
Lines 296-300: Once again, this is a very interesting finding, but it is hard to really interpret without a multiple sequence alignment or phylogenetic tree to show the relationshiips. For example, the authors describe that the DR cluster has a partial match, but that isn't an informative statement, because it is unclear what about the match is partial. This ambiguity could be cleared up by showing a multiple sequence alignment.
Have added the multiple sequence alignment for a bacteriophage from each of the clusters in Figure S4.
Lines 370-376: I do not understand this point. The authors describe that Patience and Rosebush have increased their capsid size, but compared to what? They have about the same capsid sizes when compared to each other based on the internal diameter, so they haven't increased their capsid size relative to eachother, so to which phage are the authors comparing these phages to?
We are comparing Patience and Rosebush to a typical T=7 capsid. In order to increase the size of a T=7 capsid, there are three possible mechanisms. 1: Increase the number of coat proteins used in the capsid, for example by making a T=9 capsid (Rosebush). 2: Use a minor capsid protein (Patience) while keeping the same number of capsid proteins. 3: Become prolate.
Have added the following line “In this context we are describing how a virus with T=7 capsid architecture might increase the capsid size.”.
Minor issues about the interpretability of the data by the broader community:
Results Section 3.1: This is a very minor point, but this section reiterates known data (and cites it appropriately); to some, it might appear a bit disingenuous to place this section in the Results rather than the Background.
Section 3.1 has been moved to section 2.1 in the methods.
Lines 368-369: The speculation that the decoration proteins are important for host cell interaction is intriguing, and this statement should cite other instances where such capsid associated proteins were useful in binding the host. However, I think it is also important to list other possible functions for these proteins that is also consistent with what is known about decoration proteins in phages, as not to create the idea that binding to the host is their definitive function. While the authors do say that this function is speculative, people unfamiliar with the field and less scrupulous, might interpret that statement as fact and thus propagate the idea that structural evidence shows that decoration proteins are important for host binding. Having alternative hypotheses on what these proteins might bind would help prevent such confusion.
Add the following text: “Enhancement of capsid attachment to the host has been observed with the ϕ29 bacteriophage head fibers {Xiang:2011gx}. However, it must be pointed out that decoration proteins in other bacteriophages, such as Hoc and Soc in the T4 bacteriophage, have been shown to aid in capsid stability {Sathaliyawala:2010ve}. It has also been shown that the head fibers in ϕ29 are not essential for phage infection in laboratory experiments {Salas:1972te}. Therefore, it may be that gp17 of Rosebush does have a role in capsid stability or some other unknown function.”
Lines 403-405: The authors use this paragraph how the multiplexing of phage caspids is useful for rapidly multiplexing and deciphering viral structures. They describe that at 6 angstrom resolution, the HK97 fold is decipherable and therefore useful in identifying structural groups. This information is highly useful for the field as a technical benchmark to achieve, but the information is only useful to those who study the structure of Caudovirales. However, other phages do not use this structure, such as the Cortico viridae and Tectiviridae that have a PRD1 like capsid, the Cystoviridae that are BTV-like and folds in things like Leviviridae and Inoviruses have completely novel capsid structures as well. If the authors wish to claim that multiplexing phage capsids is a successful strategy for lowering the cost per structure, a discussion of how this would be useful for other families of phages should be included, as well as a speculation about what level of resolution would be necessary to resolve them. Otherwise, this method is solely useful for Caudovirales phages, which is useful, but should be specified as such.
We think it should be applicable to most viruses. At medium resolution (5-10 angstrom) the secondary structures are visible but tracing the backbone is challenging (if not impossible). However, having visible secondary structures would allow for fold determination and differences between folds. Have added the following section:
“At the 5-10 angstrom resolution range (so called medium resolution), secondary structures, for example alpha helices, are visible, but tracing the backbone of the protein chain is very challenging {Baker:2012kx}. For most protein folds, for example the common double jelly roll fold found in PRD1 {Abrescia:2004bs}, medium resolution cryo-EM maps should have visible secondary structures within the folds and so allow for identification and classification of the viral capsids..”
Reviewer 2 Report
The manuscript detailed a comprehensive exploration on structures of three actinobacteriophage capsids. The authors used cryo-electron microscopy to obtain the high resolution structures and found some of the folds are novel. These findings may be interesting to viruses studies and show some limitations of nowadays protein modelling methodologies. The study is technical sound to me and written well.
I only have one suggestion:
I-Tasser is excellence at homology modelling, if the folds of gp29, gp17 are novel, you can try Quack, which is from the same lab but designed for de novo protein structure prediction. Or try Rosetta, a well established tool/web service successfully designed novel folds from David Becker Lab.
Author Response
Reviewer 2:
The manuscript detailed a comprehensive exploration on structures of three actinobacteriophage capsids. The authors used cryo-electron microscopy to obtain the high resolution structures and found some of the folds are novel. These findings may be interesting to viruses studies and show some limitations of nowadays protein modelling methodologies. The study is technical sound to me and written well.
I only have one suggestion:
I-Tasser is excellence at homology modelling, if the folds of gp29, gp17 are novel, you can try Quack, which is from the same lab but designed for de novo protein structure prediction. Or try Rosetta, a well established tool/web service successfully designed novel folds from David Becker Lab.
Thank you for the suggestions. Quark has a limit of 200 residues so is of limited use for these proteins. Gp17 is 249 residues. Gp29 (139 residues) was submitted. It provided a slightly different structure than I-Tasser, but it didn’t improve the fit into the cryoEM density.
Gp29 and gp17 were submitted to the Robetta web server as suggested. We are still awaiting the results.
Reviewer 3 Report
This manuscript by Podgorski et al. describes cryo-electron microscopy, with symmetric reconstruction of the DNA-containing capsid shells, of three bacteriophages that infect Mycobacterium smegmatis. The objective of the study was not stated. (We have 3,000 phages, so let’s do some structures does not state an objective.) However, an objective appears to be implicit in the structures of the accessory/ancillary proteins. The lack of a stated objective permeates the manuscript and makes reading it a chore, in spite of the fact that the basic content is quite simple. A result is a larger-than-acceptable amount of incomplete work. Much of the manuscript reads as though someone had transcribed a laboratory notebook. Thus, I think that this manuscript cannot be published as is. In theory, improvements can be made to (1) give this manuscript purpose and (2) introduce more significant conclusions. The following are major and minor details.
Major Details
[1] The first two paragraphs of the Introduction have no relationship to any discernable objective and much of this text is not relevant to the rest of the manuscript. The third paragraph starts the manuscript. However, much more detail about previous studies of accessory proteins is needed. The fourth paragraph of the Introduction can be reduced to one sentence.
[2] P2., L79: Dr. Hatfull is an author on the manuscript. Thus, you cannot thank him for phages as though he were not an author.
[3] P3, L100-101: Density is not the same as refractive index. 1.382 is about the right refractive index for phages. 1.3087 is a mistake of some kind (0 should be deleted?). Also, based on the way the phage sample was obtained, PEG in variable amount will be carried over into the cesium chloride mixture, thereby distorting the refractive index. The mixture should be reported as weight of cesium chloride per volume of resuspended phages.
[4] Sections 2.2 and 2.3 are so close to each other that they should be one section with only one iteration of what is now duplicated text. Also, the terminology of these two sections is not internally consistent.
[5] Section 2.4: Why was not the concentration of the phages determined?
[6] P 4, L168-175 do not understandably describe recognizable procedures.
[7] Section 3.1 does not belong in the Results Section. Most of the information in this section is not relevant for any recognizable objective.
[8] P5, L215-258: What did mixing of the phages achieve (in time and cost saving) if yet another sample and analysis was needed to demonstrate that the structures had been separated? I am guessing that nothing was achieved. In any case, these lines are a total distraction from any perceivable objective.
[9] P6, L231: The word, hexavalent, has no detectable meaning based on Figure 4.
[10] P6, L237-242: Mass spectrometry is not (yet) a procedure that produces accurate estimates of protein amount/stoichiometry. This text suggests that it is. If protein amount was, in fact , determined from SDSPAGE, then that should be stated. Also, what predicted the molecular weights of the proteins indicated? Cryo-EM?
[11] P6, L249: What predicted the structure of gp29? Also, nothing definitive is shown in this paragraph. One wonders whether more work should have been done.
[12] P6, L259-267: The structural features indicating an HK97-fold are stated, but not demonstrated in a figure. Then, a negative result with structure prediction is used to draw a conclusion that could have been tested with either a more complete analysis of the current structure or the obtaining of a higher resolution structure. Most of this section is not complete enough for publication. On the other hand, the differing distributions of the decoration proteins is emerging as a significant theme.
[13] Section 3.4 produces nothing definitive. The role suggested for gp17 can be tested by the dimensions of the decoration protein projected from the cryo-EM. Also, the second sentence has grammatical problems. This is another section on work that is not complete.
[14] Section 3.5 is yet another section in which the work is not complete. Also, presenting a detailed comparison to proteins in the various phages, while pointless and boring to most readers, also highlights my conclusions that, as written, this manuscript (1) has no objective and (2) is basically a transcription of either a or several laboratory notebooks.
[15] The figures should be blended with the text, not in a separate section.
[16] P14, L370-379: This is an unacceptably trivial discussion.
[17] P14, L379-383: Lambda D protein and the HK97 cross-links should have been discussed in the Introduction and the relatively definitive data on D protein’s stability promoting properties should also have been discussed in the Introduction. Then, in the Discussion Section, you can compare your protein to lambda D protein. How closely related is the positioning, for example, of the minor protein of Patience, based on the cryo-EM?
[18] P14, L384-391. Based on my comments in [12], I think that this text has no function.
[19] P14, L397-398. Where in the manuscript was this shown? I could not find it. See my Comment [8].
[20] P15, L407-411. This is an important and likely true statement. But, the current manuscript has done nothing, as far as I can tell, to advance in the direction indicated.
[21] The statement in Section 5 suggests that the authors can write a better manuscript, possibly including more extensive data.
[22] Finally, I note that work with phages is an excellent means for students to become introduced to laboratory practice. It is also a relatively rapid way to obtain data of general importance. However, from there to where this manuscript should be going, work on phages faces the same challenges that other work faces.
Minor Details
P3: The ° symbol is in the wrong place.
Figure 1: A scale bar should be provided for the cryo-EM images.
P7, L282: number agreement needed
Author Response
Reviewer 3.
This manuscript by Podgorski et al. describes cryo-electron microscopy, with symmetric reconstruction of the DNA-containing capsid shells, of three bacteriophages that infect Mycobacterium smegmatis. The objective of the study was not stated. (We have 3,000 phages, so let’s do some structures does not state an objective.) However, an objective appears to be implicit in the structures of the accessory/ancillary proteins. The lack of a stated objective permeates the manuscript and makes reading it a chore, in spite of the fact that the basic content is quite simple. A result is a larger-than-acceptable amount of incomplete work. Much of the manuscript reads as though someone had transcribed a laboratory notebook. Thus, I think that this manuscript cannot be published as is. In theory, improvements can be made to (1) give this manuscript purpose and (2) introduce more significant conclusions. The following are major and minor details.
Major Details
[1] The first two paragraphs of the Introduction have no relationship to any discernable objective and much of this text is not relevant to the rest of the manuscript. The third paragraph starts the manuscript. However, much more detail about previous studies of accessory proteins is needed. The fourth paragraph of the Introduction can be reduced to one sentence.
We appreciate the time that the reviewer has taken to read our manuscript and provide helpful feedback. Over all we feel that we have a difference in style as opposed to the substance. As the reviewer states, the “basic content is quite simple” and none of the comments by the reviewer highlight major flaws in the science that underpins the paper.
Paragraphs 1 and 2 give background to the actinobacteriophage and why the study of them is important. Once again, we appreciate the reviewer’s comments and opinion that the paper should focus more on the accessory proteins. We think that the level of detail is appropriate and that detail of the accessory proteins is best kept to the discussion.
We think it is important to highlight the work of students in isolating the actinobacteriophage, which would be lost if we remove that sentence.
[2] P2., L79: Dr. Hatfull is an author on the manuscript. Thus, you cannot thank him for phages as though he were not an author.
Changed to: “The bacteriophages Patience, Rosebush, Myrna and their host, Mycobacterium smegmatis mc2155 [10] were retrieved from the SEA-PHAGES archive at the University of Pittsburgh.”
[3] P3, L100-101: Density is not the same as refractive index. 1.382 is about the right refractive index for phages. 1.3087 is a mistake of some kind (0 should be deleted?). Also, based on the way the phage sample was obtained, PEG in variable amount will be carried over into the cesium chloride mixture, thereby distorting the refractive index. The mixture should be reported as weight of cesium chloride per volume of resuspended phages.
Changed to: “8.5g of CsCl was added to the 10 mL of bacteriophage lysate for a final density of 1.5 g CsCl/mL of bacteriophage lystate”.
[4] Sections 2.2 and 2.3 are so close to each other that they should be one section with only one iteration of what is now duplicated text. Also, the terminology of these two sections is not internally consistent.
Combined the two sections.
[5] Section 2.4: Why was not the concentration of the phages determined?
We only received enough bacteriophage lysate to carry out the cryo-EM and mass spectroscopy. Also, it is difficult to correlate spectroscopy readings with the number of bacteriophage particles. Titering is not useful for cryoEM bacteriophage samples since there is not a common correlation between number of bacteriophage particles and the number of infectious bacteriophage particles.
[6] P 4, L168-175 do not understandably describe recognizable procedures.
They are commonly used methods in mass spectroscopy.
[7] Section 3.1 does not belong in the Results Section. Most of the information in this section is not relevant for any recognizable objective.
Moved to section 2.1
[8] P5, L215-258: What did mixing of the phages achieve (in time and cost saving) if yet another sample and analysis was needed to demonstrate that the structures had been separated? I am guessing that nothing was achieved. In any case, these lines are a total distraction from any perceivable objective.
We have described a method (multiplexing) that allows for multiple phage to be added to the same grid. We believe it is of interest to the wider field. In terms of this specific experiment, there is still a cost saving. For three different bacteriophage you would need three grids. Even with the duplication we used (in order to show that chimeric structure artefacts aren’t produced, which is an important control in this proof-of-principle experiment) we only used two grids. Each grid requires one day of data collection. Three grids need three days of data collection, two grids need two days. Each day of data collection costs $3000 so there was a saving of $3000 in this experiment. We think this section should stay in the manuscript because of its wider interest in the field.
[9] P6, L231: The word, hexavalent, has no detectable meaning based on Figure 4.
At each of the six sides of the hexamer there is a minor coat protein. We therefore feel hexavalent is appropriate.
[10] P6, L237-242: Mass spectrometry is not (yet) a procedure that produces accurate estimates of protein amount/stoichiometry. This text suggests that it is. If protein amount was, in fact , determined from SDSPAGE, then that should be stated. Also, what predicted the molecular weights of the proteins indicated? Cryo-EM?
Changed to “The mass spectrometry analysis of the whole capsid identified the capsid subunit (gp23), gp15, gp4, gp29 and gp31.” to remove ambiguity.
Molecular weights were predicted using the protparam online tool and compared to SDS-PAGE bands. Changed text to: “Their SDS-PAGE analysis showed three prominent bands that matched the estimated (Protparam software {Wilkins:1999du}) molecular weights”
[11] P6, L249: What predicted the structure of gp29? Also, nothing definitive is shown in this paragraph. One wonders whether more work should have been done.
It is stated in line 246 that all three proteins had their structure predicted using the I-TASSER server. We think it is a worthwhile endeavor using the various protein structure prediction tools to see whether the predicted structures fitted the cryoEM density. The predicted structures were unfortunately unreliable based on the C and TM scores. Since using the amino acid to predict the structure is an obvious next step we wanted to report the lack of success so that others would not spend time trying to do so. Obviously, what is needed is a high resolution structure but that work is outside the scope of the manuscript.
[12] P6, L259-267: The structural features indicating an HK97-fold are stated, but not demonstrated in a figure. Then, a negative result with structure prediction is used to draw a conclusion that could have been tested with either a more complete analysis of the current structure or the obtaining of a higher resolution structure. Most of this section is not complete enough for publication. On the other hand, the differing distributions of the decoration proteins is emerging as a significant theme.
Medium resolution structures do not allow the tracing of the protein backbone with any great accuracy. Therefore it would not be useful to show the highly subjective segmentation of the major capsid protein monomer for Rosebush. We therefore decided to not show such a figure since it would not be based on good science. We say that the A domain and E loop like domains are visible and they can be seen in the pentamer of Rosebush in Figure 4. We are not convinced a further figure would be useful.
With regards to “more complete analysis” we presume the reviewer is talking about trying to segment out the monomer. As stated above, at this resolution it would be highly subjective and unreliable. We agree a high resolution structure is needed for Rosebush but that is outside the scope of this paper.
[13] Section 3.4 produces nothing definitive. The role suggested for gp17 can be tested by the dimensions of the decoration protein projected from the cryo-EM. Also, the second sentence has grammatical problems. This is another section on work that is not complete.
We agree it is not definitive but it does add more evidence that gp17 is the decoration protein. Further evidence used is the location of the gene in the genome (based on well-established synteny in bacteriophage genomes) and the predicted size of the protein. We don’t understand how the cryo-EM map at the current resolution can confirm the role of gp17. We agree that there is further interesting work to do but that would be for another manuscript.
Changed the second sentence to “The rationale being that the accessory protein should be present in relatively high amounts and produce a dark band on the gel.”
[14] Section 3.5 is yet another section in which the work is not complete. Also, presenting a detailed comparison to proteins in the various phages, while pointless and boring to most readers, also highlights my conclusions that, as written, this manuscript (1) has no objective and (2) is basically a transcription of either a or several laboratory notebooks.
We respectfully disagree. It is an obvious next step of interest to consider how widespread the decoration protein is in bacteriophage. Is it a one off that is only found in the B2 subcluster, is it only found in the acintobacteriophage family, or is it found in many different bacteriophages that infect very different hosts? We think readers will find it of interest.
[15] The figures should be blended with the text, not in a separate section.
That is the style of the journal submission.
[16] P14, L370-379: This is an unacceptably trivial discussion.
Once again we respectfully disagree. The different mechanisms bacteriophage have evolved to increase capsid size is of great interest. As we state in the text we believe that Rosebush and Patience are excellent examples of two such mechanisms.
[17] P14, L379-383: Lambda D protein and the HK97 cross-links should have been discussed in the Introduction and the relatively definitive data on D protein’s stability promoting properties should also have been discussed in the Introduction. Then, in the Discussion Section, you can compare your protein to lambda D protein. How closely related is the positioning, for example, of the minor protein of Patience, based on the cryo-EM?
We think it is better to introduce lambda D here. The resolution would limit the usefulness of the comparison of Patience and lambda D proteins. Until models can be fitted to the maps then that sort of analysis would be of limited use.
[18] P14, L384-391. Based on my comments in [12], I think that this text has no function.
We think it is important to put a T=9 capsid structure into the proper context. The number of structures for T=9, as we state, are few and are at worse resolution than the structure we describe. We used a number of different types of protein structure prediction software and none provided any structure that looks like an HK97-fold. As stated in the text, the E-loop and A-domain are visible (pentamer in Figure 4) and so it is an HK97-fold, but a novel variant that has not been structurally characterized.
[19] P14, L397-398. Where in the manuscript was this shown? I could not find it. See my Comment [8].
Please see our response to [8].
[20] P15, L407-411. This is an important and likely true statement. But, the current manuscript has done nothing, as far as I can tell, to advance in the direction indicated.
Please see our response to [8].
[21] The statement in Section 5 suggests that the authors can write a better manuscript, possibly including more extensive data.
We appreciate that the reviewer would like to see a different paper with a different focus. The reviewer does not appear to have any major issues with the science reported.
[22] Finally, I note that work with phages is an excellent means for students to become introduced to laboratory practice. It is also a relatively rapid way to obtain data of general importance. However, from there to where this manuscript should be going, work on phages faces the same challenges that other work faces.
We agree that bacteriophage are excellent for students to work with. We apologize but we can’t make sense, with respect to the manuscript, of “However, from there to where this manuscript should be going, work on phages faces the same challenges that other work faces.”
Minor Details
P3: The ° symbol is in the wrong place.
Changed line 103 degree to superscript.
Figure 1: A scale bar should be provided for the cryo-EM images.
Changed in Figure 1 and Figure 2.
P7, L282: number agreement needed
Sorry but we don’t understand what you mean. The name of the phage is Chipper1996 and it is protein 11 of that bacteriophage.
Reviewer 4 Report
Summary: Podgorski et al. present data describing the capsid structure of the virions of three bacteriophages that infect Mycobacterium smegmatis. The information is clearly presented and discussed in detail.
Phage genome can be accessed with stated GenBank accession numbers
Structural data at EMDB can be accessed with the stated accession number
Minor issues
Line 46: Do you mean virion morphology rather than capsid morphology? Line 48: Podoviridea are typically described to possess virions with short non-contractile tails Fix the degree symbols in section 2.2 and commas among x g values In the methods section, authors jump between using full-term bacteriophage and its abbreviation phage. Try to fix the text to be more consistent Line 125: What per cent of uranyl acetate was used? Section 2.4: Beginning sentences with a number rather than the spelling of that number is a bit unsightly. Line 187: Italicise Mycobacterium Line 197: about the phages Table 1: Italicise the phage family names under the header morphology Line 250 and 253: I-TASSER Figure 6: Fix genome maps to include labels (putative minor protein, etc) of discussed genes/proteins and increase the text size of numbers that label genes. It is very difficult to read in its current state Figure S1 is not mentioned in the main body of the text… Figure 7 & Tables S1, S2 &S3 – Headers should be placed on the table to describe the data being presented more clearly
Major issues
Section 2.4: A lot of this paragraph is discussing the preparation of phage for transmission electron microscopy. It would be best to relabel this section as so. Then move lines 126 to 128 to the next paragraph (section 2.5)
Recommendations to authors
Section 3.2. It would have been nice to include the dimensions of the phage virions. I could recommend a free to use software called ImageJ that is intuitive to install and use.
Author Response
Reviewer 4:
Summary: Podgorski et al. present data describing the capsid structure of the virions of three bacteriophages that infect Mycobacterium smegmatis. The information is clearly presented and discussed in detail.
Phage genome can be accessed with stated GenBank accession numbers
Structural data at EMDB can be accessed with the stated accession number
Minor issues
Line 46: Do you mean virion morphology rather than capsid morphology?
Changed to virion morphology
Line 48: Podoviridea are typically described to possess virions with short non-contractile tails
Changed “while Podoviridae lack the tail altogether and only have the tail fibers needed for attachment to the host” to “while Podoviridae have a short non-contractile tail”
Fix the degree symbols in section 2.2 and commas among x g values
Could only find one degree symbol that was not superscript. Line 107. Changed it.
Added commas to the x g values, line 98, 101, 103, 106.
In the methods section, authors jump between using full-term bacteriophage and its abbreviation phage. Try to fix the text to be more consistent
Have attempted to be more consistent. Changed all “phage” to “bacteriophage”
Line 125: What per cent of uranyl acetate was used?
1%. Added to text.
Section 2.4: Beginning sentences with a number rather than the spelling of that number is a bit unsightly.
Changed to “Twenty microliters”.
Line 187: Italicise Mycobacterium
Have changed it to be italicized.
Line 197: about the phages Table 1: Italicise the phage family names under the header morphology Changed.
Line 250 and 253: I-TASSER Figure 6: Fix genome maps to include labels (putative minor protein, etc) of discussed genes/proteins and increase the text size of numbers that label genes. It is very difficult to read in its current state
Is difficult to increase the text size of all the numbers because many of the boxes are too small to contain the increased text size. We have increased the text size of all the numbers of labelled genes. We have added extra labels for the discussed genes.
Figure S1 is not mentioned in the main body of the text…
Added to main text line 310.
Figure 7 & Tables S1, S2 &S3 – Headers should be placed on the table to describe the data being presented more clearly
Headers have been added.
Major issues
Section 2.4: A lot of this paragraph is discussing the preparation of phage for transmission electron microscopy. It would be best to relabel this section as so. Then move lines 126 to 128 to the next paragraph (section 2.5)
Changed title to: “2.4 Preparation of bacteriophages for transmission electron microscopy”
Moved lines to section 2.5
Recommendations to authors
Section 3.2. It would have been nice to include the dimensions of the phage virions. I could recommend a free to use software called ImageJ that is intuitive to install and use.
Dimensions have been added to Figure 1 legend.
Round 2
Reviewer 3 Report
In general, I do not see changes in the directions that I indicated. So, obviously, I do not approve. However, apparently, this manuscript will be published anyway. I think that this will not be good either for the journal or for the authors. For example, the first ~60% of the Abstract has none of the traditional content of an abstract. The content begins when the authors tell us that students infect Mycobacterium smegmatus; I thought that this is what phages did. The Abstract then tells us that the accessory proteins are new, but gives no information about what is new about them. The newly discovered aspects of the accessory proteins constitute practically the entire content of the manuscript. This is not good.
I have some additional comments.
[1] The Introduction tells us that 17,000 phages were isolated by 145 participating institutions, or 117 per institution. One might imagine that the SEA-PHAGE program should be expanded for improving the coverage of phages needed for phage therapy and other uses. However, is 117 phages per institution a good rate for these purposes? The answer depends on the number of person-weeks invested and the cost. So, now that the SEA-PHAGE program is being promoted for purposes beyond education, somewhere, the number of phages isolated per person-week should be estimated. Procedures and samples exist, elsewhere, such that about 50 phages per person-week are isolated.
[2] Page 4, line 139. I cannot remember phages ever being stored at -80 °C and then being used for EM after thawing. The reason is that phages tend to release DNA during thawing, which makes a messy specimen even if only a few phages release DNA. Something should be said about this.
[3] Most of the remaining manuscript is a best-forgotten blur of incomplete work. The most memorable aspect is the tandem accessory protein arrangement of Patience. How does this structure compare with what has been learned about previous accessory proteins?
Author Response
In general, I do not see changes in the directions that I indicated. So, obviously, I do not approve. However, apparently, this manuscript will be published anyway. I think that this will not be good either for the journal or for the authors. For example, the first ~60% of the Abstract has none of the traditional content of an abstract.
Abstract has been altered.
The content begins when the authors tell us that students infect Mycobacterium smegmatus; I thought that this is what phages did.
Yes, bacteriophage infect bacteria. However, there is no universal bacteriophage that infects all bacteria. Bacteriophages have specific host ranges and so it is important to state the bacterial host for the reader. It is unclear what change you want here.
The Abstract then tells us that the accessory proteins are new, but gives no information about what is new about them. The newly discovered aspects of the accessory proteins constitute practically the entire content of the manuscript. This is not good.
Abstract has been altered.
I have some additional comments.
[1] The Introduction tells us that 17,000 phages were isolated by 145 participating institutions, or 117 per institution. One might imagine that the SEA-PHAGE program should be expanded for improving the coverage of phages needed for phage therapy and other uses. However, is 117 phages per institution a good rate for these purposes? The answer depends on the number of person-weeks invested and the cost. So, now that the SEA-PHAGE program is being promoted for purposes beyond education, somewhere, the number of phages isolated per person-week should be estimated. Procedures and samples exist, elsewhere, such that about 50 phages per person-week are isolated.
Sorry but we are not sure how this is relevant to the paper.
[2] Page 4, line 139. I cannot remember phages ever being stored at -80 °C and then being used for EM after thawing. The reason is that phages tend to release DNA during thawing, which makes a messy specimen even if only a few phages release DNA. Something should be said about this.
Yes we agree that can happen. You can see some of the DNA strands in Figure 2 from released bacteriophage capsids in the background. We did observe some empty capsids (you can see them as a separate class in Figure S2 on the third row, second column). We don’t think the DNA had any adverse effect on the cryoEM. Even if some DNA overlaps the bacteriophage images, then it is averaged out during the refinement. The resolution (in this paper) is limited by the number of bacteriophage particles, not the presence of DNA. We do find empty capsids in almost all bacteriophage preparations, whether they have been frozen or not, so we are not sure whether the -80oC increases DNA release for all bacteriophage. The majority of capsids in the cryo grids still had DNA in the capsids so these specific bacteriophages appear to be more resistant to DNA release from thawing. The bacteriophages are stored with DMSO (6% final concentration) which has been added to the methods section.
We have added the following to the manuscript (Lines 773-775):
“We did observe a minority of bacteriophage particles had released their DNA (Figure S2, third row, second column, shows the empty particles). This may be a result of thawing the bacteriophages from -80oC.”
[3] Most of the remaining manuscript is a best-forgotten blur of incomplete work. The most memorable aspect is the tandem accessory protein arrangement of Patience. How does this structure compare with what has been learned about previous accessory proteins?
We agree it is very interesting. We have been unable to find any such example in the literature. We have added some discussion about BPP-1 and epsilon, the only other bacteriophage we have been able to find with accessory proteins that exist as dimers.
Added lines 1111-1116: “The decoration protein gp15 appears to exist as a dimer and the linking of two minor coat proteins with a decoration protein has not been observed in other bacteriophage capsids. The closest comparisons are the accessory proteins of the Bordetella infecting bacteriophage, BPP-1{Zhang:2013kx} and epsilon15{Baker:2013kw} which also exist as dimers in the capsid protein. However, in those cases the dimers are located in similar positions to minor capsid proteins and interact with the major capsid protein directly.”